# Is the Secret in the Gut? SuperJump Activity Improves Bone Remodeling and Glucose Homeostasis by GLP-1 and GIP Peptides in Eumenorrheic Women

**DOI:** 10.3390/biology11020296

**Published:** 2022-02-11

**Authors:** Sonya Vasto, Alessandra Amato, Patrizia Proia, Sara Baldassano

**Affiliations:** 1Department of Biological, Chemical and Pharmaceutical Sciences and Technologies (STEBICEF), University of Palermo, Viale delle Scienze, 90128 Palermo, Italy; sonya.vasto@unipa.it; 2Sport and Exercise Sciences Research Unit, Department of Psychological, Pedagogical and Educational Sciences, University of Palermo, 90128 Palermo, Italy; alessandra.amato02@unipa.it

**Keywords:** biological mechanisms, physical health, sports and exercise physiology, glucagon-like peptide-1, glucose-dependent insulinotropic polypeptide

## Abstract

**Simple Summary:**

We previously showed that SuperJump activity, an innovative workout training performed on an elastic minitrampoline, exerts osteogenic action in women. The present study analyzed whether the gut peptides (GLP-1, GIP, GLP-2, PYY, ghrelin) are involved in the mechanism of action. This is because there is a link between gut peptides and bone. In fact, ingestion of a meal induces secretion of the gut peptides that act by decreasing bone resorption and blood glucose level. After 20 weeks of SuperJump activity GLP-1 and GIP levels were significantly increased while fasting insulin, glucose, insulin resistance, were significantly reduced. The study suggests that GLP-1, and GIP are involved in the mechanism of action that improves bone health and blood glucose level following 20 weeks of SuperJump activity in women.

**Abstract:**

We showed that twenty weeks of SuperJump activity, an innovative workout training performed on an elastic minitrampoline, reduced bone resorption and increased bone formation in eumenorrheic women acting on the key points of the regulation of bone metabolism. The present study analyzed whether the gastrointestinal hormones are involved in the mechanism of action and if it has an impact on glucose homeostasis. The control group was composed of twelve women, similar to the exercise group that performed SuperJump activity for twenty weeks. The analysis was performed on blood samples and investigated GLP-1, GIP, GLP-2, PYY, ghrelin, glucose, insulin, insulin resistance, β-cell function, and insulin sensitivity. The results showed that the activity contributes to raising the GLP-1and GIP levels, and not on GLP-2, PYY, and ghrelin, which did not change. Moreover, SuperJump activity significantly reduced fasting insulin, glucose, insulin resistance, and increased insulin sensitivity but did not affect beta cell function. These data suggest that GLP-1, and GIP are involved in the mechanism of action that improves bone and glucose homeostasis following 20 weeks of SuperJump activity in eumenorrheic women.

## 1. Introduction

The gastrointestinal tract is the body’s largest endocrine organ secreting hormones which in turn regulate whole-body homeostasis. Therefore, many dysmetabolic conditions such as insulin resistance or higher risk of fractures are accompanied by altered secretion of gut peptides [1,2]. Gastrointestinal secretion of gut peptides is stimulated by nutrients when these reach the intestinal L cells, but levels of the gut hormones seem to be influenced by an exercise bout [3,4,5,6], suggesting that physical exercise could modulate gut peptides release.

There is a high link between gut peptides and bone. In fact, ingestion of a meal induces secretion of gut peptides that act by decreasing bone resorption [7]. The responsible gut peptides appear to include glucagon-like peptide-1 (GLP-1), glucose-dependent insulinotropic polypeptide (GIP), glucagon-like peptide-2 (GLP-2), peptide YY (PYY), and ghrelin [8]. GLP-1 and GIP are also known as incretin hormones due to their role in regulating glucose homeostasis by acting on insulin release. However, other gut peptides such as GLP-2 and PYY influence glucose metabolism [9,10,11,12,13].

Physical exercise is indispensable to improve bone health [14,15] and glucose metabolism [16,17]. It can even replace glucose-lowering medication [3]. However, thus far, how exercise improves glucose homeostasis in humans is not fully understood and the influence of exercise on beta cell adaptations remains to be clarified.

SuperJump is an innovative activity performed on an elastic minitrampoline that can be used to be fit, maintain well-being, and counteract a sedentary lifestyle due to home confinement such as during COVID-19 [18]. We have previously shown that 20 weeks of SuperJump training reduced bone resorption and increased bone formation in eumenorrheic women acting on the key points of the regulation of bone metabolism [19]. In this manuscript, it was hypothesized that gastrointestinal hormones are involved in the metabolic pathway underlying bone remodeling following SuperJump exercise in eumenorrheic women. Furthermore, since gastrointestinal hormones impact on glucose metabolism, it was secondarily hypothesized that SuperJump may have effects on glucose homeostasis and beta cell function. Therefore, the aim of the study was to investigate whether the gastrointestinal hormones, and specifically GLP-1, GIP, GLP-2, PYY, and ghrelin, are involved in the mechanism of action that influences bone remodeling following 20 weeks of SuperJump activity and whether these changes would also impact on glucose homeostasis.

## 2. Materials and Methods

### 2.1. Subjects and Experimental Design 

This study is part of a larger project (TRAMP2021). As previously described in [19], from an initial number of forty-two women, due to lack of inclusion criteria or withdrawal, twenty-four eumenorrheic women were randomized into two groups, the exercise group and the non-exercise (control) group for a total of twelve women in each group. Inclusion and exclusion criteria are summarized in Table 1. Briefly, during the first visit, the participants underwent anthropometric measurement and completed a habitual dietary intake assessment [20]. Before starting the activity, a blood sample (BASE) was collected; the second sample of blood was collected at the end of the 20 weeks (W20) (Figure 1).

#### 2.1.1. Workout Characteristic

The exercise group performed SuperJump training (CoalSport, Rome, Italy). The intensity was 65–75% HR max and the frequency was three times a week for a total of 20 weeks. The session time was 60 min. The SuperJump training session was performed by the whole exercise group on the same days, at the same time, by the exercise group together. The training sessions were carried out on the mini trampoline and led by experienced instructors. Each session was divided into five min warm-up, a central phase with full body jumping exercises, and five min cool-down phase. The central phase was a circuit of 10 exercises, 50 s each, with 10 s of active recovery each time. The circuit was repeated five times per training session. The training session was entirely performed on the mini trampoline including the recovery phase during which the subjects continued to jump on the trampoline at the minimum intensity that allowed them to perform the jump (just lift both feet off the trampoline together). The ten resistance exercises were: (1) isometric lateral raises; (2) curl, 3) oblique; (4) adductors/ abductor; (5) triceps; (6) front lifts; (7) split jump alternating drill; (8) Pull to the chin; (9) jumping jack single arm; (10) standing Russian twist. All exercises were performed with dumbbells, with the weight allowing the subject to carry out the exercise for 50 s. SuperJump is a moderate-to-vigorous activity (sRPE = 3.1 ± 1.2); during the training session, the subjects spend 47.1 ± 34.4% of the time on moderate intensity (64 ± 76.9 % of HRmax) and 34.6 ± 39.6% of the session time on vigorous intensity (77 ± 95.9 %of HRmax) [18]. The performance and the effects of SuperJump were recently studied [18,19] and the characteristics of this activity, which include resistance exercise and impact activities such as during the active recovery phase, classify it among activities with osteogenic potential. The rationale of the SuperJump protocol was to undertake both resistance exercises and impact activities to exert osteogenic effects. Indeed, it has been demonstrated that resistance exercise has a better osteogenic potential than just aerobic exercise [8]. The active recovery phase was important to prolong “the impact stimulus” of the activity over time. The impact of a physical exercise is the combination of force magnitude and the speed at which the force is applied [21]. Activities with the most osteogenic potential have ground reaction forces (GRF) greater than 3.5 times BW (per leg), with peak force occurring in less than 0.1 s [22]. Comparing three main activities such as walking, running, jumping, the last has the greatest benefits to bone mineralization [23,24]. It also seems important that not only the characteristic of the movement but also the number of repetitions, in fact 50 jumps in a session [25], does not seem to have an osteogenic effect compared to 100 jumps [26,27]. The control group did not perform physical activity during the time of the study. Physical activity was intended as structured activity and excluded daily life activities (e.g., physically heavy work) and journeys on foot or by bike to go to work.

#### 2.1.2. Anthropometry

Body composition, specifically lean mass and fat mass, was measured by electrical bioimpedance measurements (InBody 320 Body Composition Analyzer). Body weight and barefoot standing height were measured by using an electronic scale and a wall-mounted stadiometer, respectively (Gima 27335 and Gima 27088, Italy). Body mass index (BMI) was reported as weight (kilograms) per standing height (meters squared).

### 2.2. Blood Sample Collection

Blood samples were collected by a specialist the morning after overnight fasting. For plasma samples, we used a tube containing EDTA while for serum, we allowed it to clot in serum tubes at room temperature for 30 min before being centrifuged under the same conditions and were centrifuged at 3000 rpm for 10 min.

### 2.3. Assays

To measure gut peptides, plasma samples were collected in pre-chilled EDTA-containing tubes with apoprotein (0.6 TIU/mL blood) and dipeptidyl peptidase IV inhibitor (10 µL/mL blood). Plasma was obtained by centrifugation at 5 °C for 10 min at 3000 rpm for measurements of GLP-1, GIP, ghrelin, PYY, and GLP-2. All samples were immediately stored at −80 °C until analyzed. All samples were analyzed in duplicate. As previously reported [20], human plasma peptide samples were analyzed using the following enzyme immunoassay kit: EZGLPHS-35K for active GLP-1, EZHGIP-54K for total GIP, EZGRT-89K for total ghrelin, EZHPPYYT66K for total PYY, and EZGLP-237K for GLP-2, all from Millipore. The inter- and intra-assay coefficients of variation for total ghrelin were 6.62% and 1.32%; 6.62% and 5.15% for GLP-2; 11.5% and 4.5% for active GLP-1; 7.41% and 2.27% for total PYY; and 3.37% and 6.45% for total GIP. All samples were measured in one assay to avoid inter-assay variation. Glucose, insulin, total cholesterol, HDL-cholesterol, and triglycerides were measured by standard commercial assays supplied by Roche Diagnostics performed on the Roche COBAS c501. The HOMA2 computer model was used to estimate insulin resistance (HOMA2-IR), β-cell function (HOMA2-%B), and insulin sensitivity (HOMA2-%S) from fasting insulin and glucose concentrations calculated by the HOMA2 calculator for specific insulin version 2.2.3, available from http://www.dtu.ox.ac.uk/homacalculator, accessed on 28 November 2021. The method is an updated HOMA model and has been used extensively to measure insulin resistance β-cell function and insulin sensitivity [28,29].

### 2.4. Ethics

The study conducted in accordance with the Declaration of Helsinki was approved by ethics committee 1 of the University of Palermo, Policlinico Giaccone Hospital, approval number 2–2020–27. Before the start of the study, all subjects involved provided written informed consent. In addition, the clinical study was registered on Clinicaltrials.gov under number NCT04942691.

### 2.5. Statistics

Based on the results of previous studies on exercise and gut peptides [3,30], the study was powered to detect a change in GLP-1 of 30% (SD 20%) considering a Type I error (α) = 0.05 (two-sided), and Type II error (β) = 0.20 (power of 80%). An a priori power calculation determined that ten subjects were required to achieve 80% power at *p* < 0.05 by using G Power software. The comparison between the groups was performed by one-way ANOVA followed by Tukey’s posttest. A *p* < 0.05 was considered to be statistically significant by using GraphPad Prism software.

## 3. Results

The cohort under investigation did not show significant differences in body mass index or composition among the groups (control vs. exercise) or within the groups (time zero vs. 20 weeks). Instead, a significant difference was observed in triglyceride levels in the exercise group at W20 compared with BASE, while no differences were reported in total cholesterol, HDL-cholesterol, and LDL-cholesterol between the groups (control vs. exercise) or within the groups (time zero vs. 20 weeks) (Table 2). Moreover, we previously showed that there was significant change in the markers of bone remodeling after 20 weeks of training in the exercise group. The marker of bone formation, osteocalcin, increased from 16.2 ± 5 (BASE) to 22.2 ± 6 μg/L (W20). The marker of bone resorption, CTX, decreased from 0.44 ± 0.1 (BASE) to 0.29 ± 0.1 μg/L (W20). PTH decreased from 44 ± 15 (BASE) to 34 ± 11 ng/L (W20). Calcitonin, vitamin D, and phosphate concentrations did not change while there was a significant increase in calcium and potassium concentrations [19].

### 3.1. Incretins

In the control group, there were no significant changes in plasma GLP-1 and GIP levels at W20 compared with BASE within the group (Figure 2). In the exercise group, GLP-1 was significantly increased at W20 compared with BASE (Figure 2A). GLP-1 concentrations at W20 increased by 58% from BASE (5.7 ± 1.7 vs. 3.6 ± 0.7 pmol/L). The levels detected were within the normal range. In addition, the GIP level was significantly increased. Specifically, in the exercise group, GIP increased by 102% at W20 compared to BASE (64.3 ± 21 vs. 31.9 ± 12 pg/mL) (Figure 2B) and the concentrations detected were within the physiological range. There was a significant change in the endogenous levels of GLP-1 and GIP in the exercise group at W20 compared to the control group (Figure 2A,B).

### 3.2. Other Gut Hormones

In the control group, there were no changes in GLP-2 (2.1 ± 0.25 vs. 2 ± 0.26 ng/mL) PYY (58 ± 11 vs. 56 ± 12 pg/mL) and ghrelin levels (1081 ± 491 vs. 1036 ± 376 pg/mL) at W20 compared with BASE. Moreover, in the exercise group, SuperJump training did not affect plasma GLP-2 (2.1 ± 0.20 vs. 1.9 ± 0.16 ng/mL), PYY (55 ± 11 vs. 52 ± 6 pg/mL), and ghrelin concentrations (947 ± 351 vs. 1014 ± 458 pg/mL) at W20 compared to the baseline. Additionally, the comparison between the two groups (control vs. exercises) showed no significant changes in GLP-2, PYY, and ghrelin (Figure 3A–C).

### 3.3. Markers of Glucose Homeostasis

In the control group, there was no difference in fasting glucose (88 ± 3.1 vs. 90 ± 4.2 mg/dL), insulin (8.1 ± 2.8 vs. 8.6± 3.0 mUI/L), or insulin resistance (1.0 ± 0.4 vs. 1.2 ± 0.3) at W20 compared with BASE. In the exercise group, SuperJump training significantly reduced fasting glucose (80 ± 7.4 vs. 88 ± 4.0 mg/dL), insulin (4.7 ± 1.9 vs. 7.9 ± 2.6 mUI/L), and insulin resistance (0.6 ± 0.2 vs. 1.0± 0.3) at W20 compared with BASE. The comparison between the groups (control vs. exercise) showed a significant reduction in fasting insulin, glucose, and insulin resistance in the exercise group at W20 compared to the control group (Figure 4A–C). There was no difference in β-cell function in the control group or the exercise group at W20 compared with BASE (Figure 4D). There was no difference in insulin sensitivity in the control group while in the exercise group, there was a significant increase in insulin sensitivity at W20 compared with BASE. The comparison between the groups (control vs. exercises) showed a significant increase in insulin sensitivity at W20 in the exercise group compared to the control group (Figure 4E).

## 4. Discussion

In previous studies, the endogenous levels of GLP-1 and GIP following physical activity have been measured at the end of the single training session [2] and have not investigated the potential link between gut peptides, bone remodeling, and physical activity.

This study shows that the gut peptides GLP-1 and GIP are involved in the mechanism of action that influences bone remodeling and ameliorates glucose homeostasis following 20 weeks of SuperJump training in eumenorrheic women.

We previously showed that SuperJump activity exerts osteogenic action in eumenorrheic women. In fact, after 20 weeks of SuperJump training, the levels of the marker of bone resorption CTX were significantly reduced while the levels of the marker of bone formation osteocalcin were increased. We found that PTH, calcium, and potassium were involved in the mechanism of action [19]. The present study showed that the SuperJump exercise program for 20 weeks significantly increased endogenous GLP-1 and GIP levels, suggesting that these two incretins are part of the mechanism of action by which this type of high impact activity influences bone remodeling in eumenorrheic women. This was confirmed by the lack of changes in the endogenous level of GLP-1 or GIP in the control group of sedentary women. We observed an increase in the endogenous levels of GLP-1 and GIP that was positive for bone remodeling because it is within the normal physiological range. In fact, in women, treatment with the long-acting agonist of the GLP-1R, liraglutide, increased P1NP (bone formation marker) and bone mineral content and reduced the bone loss, indicating that GLP-1 acted by increasing bone formation [31]. In ovariectomized rats, the treatment with liraglutide increased bone mineral density and improved trabecular thickness, number, and volume [32]. Moreover, the activation of GLP-1R decreased P1NP secretion and increased cell viability in osteoblasts [33]. Additionally, GIP exerts an anti-resorptive action and anabolic effect [34]. GIP stimulated the expression of P1NP and of ALP activity [35] and reduced the level of CTX, the marker of bone resorption [36]. The GIP receptor is expressed in osteoblast and osteoclast derived cell lines. Therefore, the loss of function for the GIP receptor gene, in women carrying the gene polymorphism E354Q, was correlated with decreased bone mineral density and increased risk of fractures [37].

Regarding GLP-2, previous studies have shown that GLP-2 administered subcutaneously in postmenopausal women reduced CTX, markers of bone resorption, and had a minimal effect on bone formation [38]. In our study, GLP-2 levels did not differ between the exercise and control groups, suggesting that it is not involved in the mechanism of action that impacts on bone remodeling in exercising women. We cannot exclude that 20 weeks of SuperJump training were not sufficient to induce differences in the endogenous levels of the peptide. However, on the basis of previous studies, supraphysiological doses of exogenous GLP-2 are necessary to reduce bone resorption [1]. Thus, changes within the physiological range may not be sufficient to see an effect and this may account for the lack of differences. However, thus far, it is still unknown whether GLP-2 affects bone metabolism directly or indirectly by involving other intestinal factors. In fact, the GLP-2 receptor has not been identified in human osteoclasts or in any other bone-related cell types [34].

Evidence from human studies indicates that PYY modulates bone homeostasis [1]. The PYY increases were associated with low bone mineral density in women with weight alteration [39,40] and absence of menstrual periods [41]. In our study, we did not find any difference in PYY concentration, ruling out an involvement of PYY. This is probably because our study population was constituted of eumenorrheic women with no weight alteration. We did not also find any differences in circulating ghrelin levels in the groups of study. Ghrelin is first a regulator of energy metabolism but seems to influence bone [42]. However, the basal concentration of ghrelin is inversely associated with body mass index. In fact, reduced ghrelin levels were found in obese people [43].

Thus far, we know that physical activity ameliorates glucose homeostasis, but it is still unclear how it acts to do so. Here, we suggest that GLP-1 and GIP could be part of the physiological mechanism of action that improves glucose homeostasis following a high impact physical activity. In fact, the higher endogenous GLP-1 and GIP level in the exercise group following 20 weeks of SuperJump activity improved glucose metabolism. In the exercise group, reduced fasting glucose, insulin and insulin resistance, and increased insulin sensitivity was observed. This was confirmed by the lack of changes in fasting glucose, insulin, or insulin sensitivity in the control group of the study. This agrees with the reduced gut peptide responses reported in sedentary obese people that develop insulin resistance [44]. The observation of elevated GLP-1 and reduced insulin could be counterintuitive in consideration of the ability of GLP-1 to stimulate insulin release. Thus, it is necessary to point out that the incretin effect is defined as the increase in insulin response after an oral ingestion of glucose. In fact, GLP-1 induces insulin secretion via the GLP-1R in a glucose-regulated manner [45]. The blood samples in the groups of study were obtained after an overnight fast. Thus, it is possible to hypothesize that fasting levels of insulin and glucose were lower thanks to a regulatory mechanism of the peptide on beta-cells that could be sensitized to secrete the minimum amount of insulin required to have an accurate glycemic control. In fact, insulin sensitivity was increased. Further studies are required.

A key strength of the present study was to analyze the effects of chronic exercise (20 weeks of SuperJump exercise) on endogenous peptides with respect to previous studies that have focused on the effects of acute exercise on the secretion of gastrointestinal hormones [46]. In fact, to our knowledge, the endogenous levels of GLP-1 and GIP following physical activity have only been measured acutely, at the end of the training session [2,46,47,48,49,50,51], and not after several weeks of training program like in our study. Moreover, the mechanism of action and therefore the potential link between gut peptides, bone remodeling, glucose metabolism, and physical activity has not been investigated. For GLP-1, the studies measured total and not the active form of GLP-1 l such as in our study [51]. However, GLP-1, similar to us, showed an increase in basal GLP-1 levels [46,51,52]. These investigations were conducted after acute exercise not only in normal weight, but also in obese trained women and suggest that endogenous levels of GLP-1 are very sensitive to physical activity. For GIP, the studies have been conducted in obese/diabetic cohort of patients or following a glucose tolerance test [53,54,55]. These studies were unconcluded and showed a decrease, increase, or no change in the GIP concentrations. We are conscious that we did not compare the SuperJump group with another group that performed other forms of exercise such as a different high impact or strength training. Thus, we do not know the effects of other forms of chronic exercise on basal gut peptide release and further studies are necessary. In fact, the exercise protocol characteristics such as the age, fitness level, BMI, and the exercise protocols such as duration and intensity could differently impact on gut peptide release. which is a limitation of the study. We also do not know whether the observed effects were mediated by GLP-1 or GIP. We may suppose that the observed effects are mediated by a synergistic action of the two peptides, but further studies are necessary to clarify the point.

## 5. Conclusions

In conclusion, the study points out the ability of physical activity by increasing endogenous GLP-1 and GIP levels to ameliorate bone and glucose metabolism, suggesting that the peptides are involved in the physiological mechanism of action that improves bone and glucose homeostasis following 20 weeks of SuperJump activity in eumenorrheic women.

## Figures and Tables

**Figure 1 biology-11-00296-f001:**
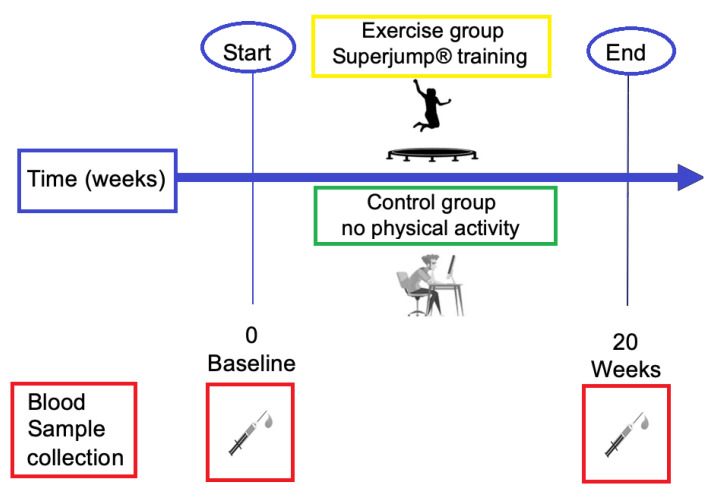
Overview of the experimental design. Blood samples were collected at baseline, at time 0, and after twenty weeks in the two groups of study (control group and exercise group). In the exercise group, SuperJump activity was performed for a total of 20-weeks, three times a week, 60 min each session. The control group did not perform physical activity.

**Figure 2 biology-11-00296-f002:**
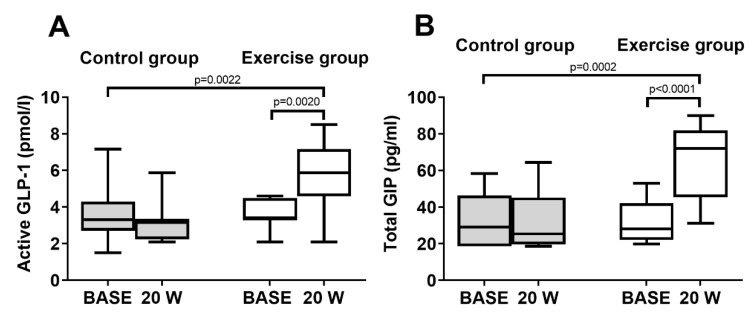
Endogenous incretin levels measured baseline (BASE) and after 20 weeks (W20) in the control group and exercise group. (**A**) Box and whisker plot of GLP-1. (**B**) Box and whisker plot of GIP.

**Figure 3 biology-11-00296-f003:**
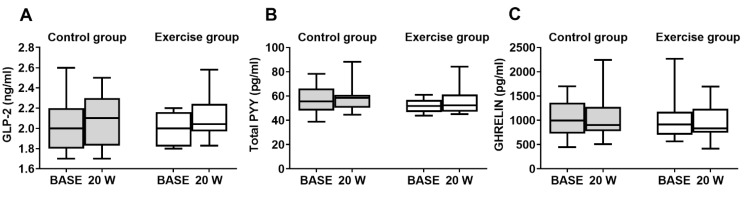
Endogenous gut peptide levels measured at baseline (BASE) and after 20 weeks (W20) in the control group and exercise group. (**A**) Box and whisker plot of GLP-2. (**B**) Box and whisker plot of PYY. (**C**) Box and whisker plot of ghrelin.

**Figure 4 biology-11-00296-f004:**
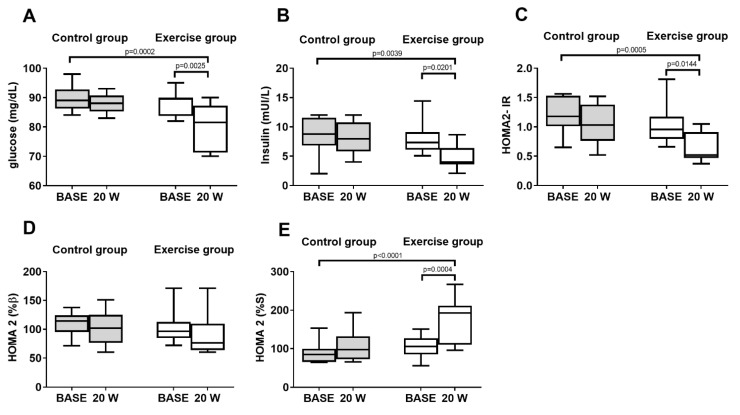
Markers of glucose homeostasis measured at baseline (BASE) and after 20 weeks (W20) in the control group and exercise group. (**A**) Box and whisker plot of fasting glucose. (**B**) Box and whisker plot of fasting insulin. (**C**) Box and whisker plot of insulin resistance. (**D**) Box and whisker plot of β-cell function. (**E**) Box and whisker plot of insulin sensitivity.

**Table 1 biology-11-00296-t001:** Inclusion and exclusion criteria of the study.

Inclusion Criteria	Exclusion Criteria
Women living in Italy	Bone fracture within the previous year
Age: 18–40 years	Self-reported long (>35 days) or short (<24 days) or irregular menstrual cycles
Currently injury free	Use of medication or suffering from any condition known to affect bone metabolism
Body mass index between 18.5 and 28 kg/m^2^	Pregnancy, breastfeeding
Menstrual cycle interval between 24 and 35 days	Current smokers
	Use of any type of hormonal contraception within the past six months
	Calcium or vitamin D supplementation in the preceding six months
	Participation in moderate and high impact-activity for ≥3 h·week before enrolling in the study

**Table 2 biology-11-00296-t002:** Characteristics of the subjects measured baseline and after 20 weeks (20 W) in the two groups of women.

Subjects Charact	Control Group			Exercise Group		
	BASE	20 W		BASE	20 W	
	Mean ± SD	Mean ± SD	*p*-Value	Mean ± SD	Mean ± SD	*p*-Value
**BMI (kg/m^2^)**	22.5 ± 2.7	23.7 ± 2.9	*p* > 0.05	22.8 ± 2.4	22.8 ± 2.8	*p* > 0.05
**LM %**	74.4 ± 5.8	76.6 ± 5.2	*p* > 0.05	73.2 ± 5.9	73.7 ± 7.2	*p* > 0.05
**FM %**	25.6 ± 5.8	23.9 ± 6.2	*p* > 0.05	26.8 ± 6	26.3 ± 7.2	*p* > 0.05
**TRIG (mg/dL)**	91 ± 14	89 ± 26	*p* > 0.05	80 ± 21	55 ±18	*p* = 0.02
**Total Chol (mg/dL)**	182 ± 18	188 ± 27	*p* > 0.05	182 ± 23	184 ± 24	*p* > 0.05
**HDL-Chol (mg/dL)**	77 ± 12	75 ± 11	*p* > 0.05	77 ± 15	80 ± 14	*p* > 0.05
**LDL-Chol**	93 ± 24	98 ± 18	*p* > 0.05	100 ± 14	89 ± 16	*p* > 0.05

**Abbreviation**: Charact, Characteristics; BMI, Body Mass Index; LM, Lean Mass; FM, Fat Mass; TRIG, Triglycerides; Chol, Cholesterol.

## Data Availability

The datasets during and/or analyzed during the current study are available from the corresponding author on reasonable request.

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
