# Peer review of "Is the Secret in the Gut? SuperJump Activity Improves Bone Remodeling and Glucose Homeostasis by GLP-1 and GIP Peptides in Eumenorrheic Women"

_biology, 2022, doi:10.3390/biology11020296_

Round 1
Reviewer 1 Report
The manuscript brings an interesting idea about glucose metabolism and its influence on bone metabolism. However, there are flaws in the methods, and basic information about the results is lacking.
1. The authors conclude that intervention can prevent osteoporosis. However, it brings little data about bone metabolism in its results.
2. It concludes that exercise can prevent diabetes. What would be the novelty presented?
3. How was the training control carried out? Was the intervention performed at home, in a group? Was there control of frequency, duration, intensity? It is necessary to specify the intervention further.
4. How was the analysis of body composition performed? It contains data on body composition but does not present the methods.
5. Why don't they show LDL cholesterol data?
6. Are participants all healthy? Why do they conclude protection against the risk of osteoporosis if they do not assess anything specific?
Author Response
The manuscript brings an interesting idea about glucose metabolism and its influence on bone metabolism. However, there are flaws in the methods, and basic information about the results is lacking.
We would like to thank the referee for the valuable suggestions to improve the clarity of the manuscript.
- The authors conclude that intervention can prevent osteoporosis. However, it brings little data about bone metabolism in its results.
Yes, you are right. The sentence has been removed from the simple summary (lines 19-20), the abstract (lines 35-36) and the conclusion that has been changed (lines 348-350). To further improve the clarity of the manuscript we added in the results section the data about the effects of SuperJump on Bone metabolism (lines 193-200).
- It concludes that exercise can prevent diabetes. What would be the novelty presented?
We know that physical activity ameliorates glucose homeostasis but it is still unclear how physical activity acts to do it. The novelty of the manuscript is to investigate if gut peptides like GLP-1, GIP GLP-2, PYY ghrelin, are part of the mechanism of action by which physical activity improves glucose homeostasis. We have better pointed out the novelty of the manuscript (lines 305-308) and we would like to thank you for the suggestion.
How was the training control carried out? Was the intervention performed at home, in a group? Was there control of frequency, duration, intensity? It is necessary to specify the intervention further.
As you suggested we have added a new paragraph to better specify the intervention protocol (lines 104-136). We also clarified that the intervention was performed at the gym and in group sessions. The intensity was 65-75% HR max and the frequency was 3 times a week for a total of 20 weeks. The session time was 60 minutes. During the training session the subjects spend the47.1 ± 34.4% of the time on moderate intensity (64±76,9 % of HRmax) and 34.6 ± 39.6% of the session time on vigorous intensity (77±95,9 %of HRmax). The control group did not perform physical activity during the time of the study. Physical activity was intended as structured activity, and excluded daily life activities (like physically heavy work) and journeys on foot or by bike to go to work.
How was the analysis of body composition performed? It contains data on body composition but does not present the methods.
We apologize for it. Body composition (Lean mass, Fat mass) was measured by electrical bioimpedance measurement (InBody320 Body Composition Analyzer). We have added a new paragraph with the anthropometric methods (lines 139-143).
Why don't they show LDL cholesterol data?
As you required we have added the LDL cholesterol data in table 2.
- Are participants all healthy? Why do they conclude protection against the risk of osteoporosis if they do not assess anything specific?
Yes, the participants were healthy eumenorrheic women. You are right. The sentence was removed from the conclusions.
Reviewer 2 Report
Reviewer's comments on the manuscript entitled “Is the secret in the gut? SuperJump activity improves bone remodeling and glucose homeostasis by GLP-1 and GIP peptides in eumenorrheic women” (manuscript ID: biology-1504557). The aim of the study wasto investigate whether the gastrointestinal 70 hormones and specifically GLP-1, GIP, GLP-2, PYY and ghrelin are involved in the mech-71 anism of action that influences bone remodeling following 20 weeks of SuperJump activity 72 and whether these changes would also impact on glucose homeostasis. The work is properly prepared and therefore, I only have a few comments and suggestions:
- In the abstract, the authors of the study should remove the names of individual parts.
- The authors should improve the manner of indicating numbers of citation so that they are in accordance with the journal guidelines for authors.
- In my opinion, Figure 1 is redundant. However, the authors are free to decide whether to remove or leave it.
- Line 99: The date must be written in a correct format.
- What was the speed of centrifugation to obtain blood serum?
- I have not found an approval number of the Bioethics Committee.
- In Table 2, I suggest correcting the names in lines as follows, for instance: replace "cholesterol" into "total cholesterol", etc.
- In subsection 3.2., the authors of the study should take the mean and standard deviation for the analysed hormones into consideration. The same applies to subsection 3.3. for markers of glucose homeostasis.
- The discussion is interesting and straight-to-the point and therefore, I have no suggestions as to this part of the manuscript.
- There are no strengths and weaknesses of the study. Could the authors complete this?
- It is necessary to improve the conclusions as they should refer to the research findings.
Author Response
We would like to thank the referee very much for the comments to further improve the manuscript.
Reviewer's comments on the manuscript entitled “Is the secret in the gut? SuperJump activity improves bone remodeling and glucose homeostasis by GLP-1 and GIP peptides in eumenorrheic women” (manuscript ID: biology-1504557). The aim of the study was to investigate whether the gastrointestinal hormones and specifically GLP-1, GIP, GLP-2, PYY and ghrelin are involved in the mechanism of action that influences bone remodeling following 20 weeks of SuperJump activity and whether these changes would also impact on glucose homeostasis. The work is properly prepared and therefore, I only have a few comments and suggestions:
- In the abstract, the authors of the study should remove the names of individual parts.
Done.
- The authors should improve the manner of indicating numbers of citation so that they are in accordance with the journal guidelines for authors.
The reference style has been changed according to the journal guidelines for authors.
- In my opinion, Figure 1 is redundant. However, the authors are free to decide whether to remove or leave it. Thank you for the suggestion. We decided to leave the figure 1.
- Line 99: The date must be written in a correct format.
done
- What was the speed of centrifugation to obtain blood serum?
The speed was 3000 rpm. The information has been added in line 150.
- I have not found an approval number of the Bioethics Committee.
- The approbation number is specified in line 175.
- In Table 2, I suggest correcting the names in lines as follows, for instance: replace "cholesterol" into "total cholesterol", etc.
done
- In subsection 3.2., the authors of the study should take the mean and standard deviation for the analyzed hormones into consideration. The same applies to subsection 3.3. for markers of glucose homeostasis.
done
- The discussion is interesting and straight-to-the point and therefore, I have no suggestions as to this part of the manuscript.
- There are no strengths and weaknesses of the study. Could the authors complete this?
As you suggested these points has been added in the discussion (lines 323-345). A key strength of the present study was to analyze the effects of chronic exercise (20 weeks of Superjump exercise) on endogenous peptides while previous studies have focused on the effects of acute exercise on secretion of gastrointestinal hormones. Moreover, For GLP-1 the studies measured total and not the active form of GLP-1 like in our study. For GIP the studies have been conducted in obese/diabetic cohort of patients or following a glucose tolerance test. We are conscious that we did not compare the Superjump group with another group that performed other forms of exercise like a different high impact or strength training. Thus, we don’t know the effects of other forms of chronic exercise on basal gut peptide release and further studies are necessary. We also don’t know if the observed effects were mediated by GLP-1 or GIP.
- It is necessary to improve the conclusions as they should refer to the research findings
We have changed the conclusion to refer to the research findings (lines 348-350) by pointing out the ability of physical activity by increasing endogenous GLP-1 and GIP levels to ameliorate bone and glucose metabolism suggesting that the peptides are involved in the physiological mechanism of action that improves bone and glucose homeostasis following 20 weeks of Superjump activity in eumenorrheic women.
Reviewer 3 Report
A link between exercise and gut peptide production is well established. How does the SuperJump protocol differ from other forms of exercise in its ability to induce gut peptide production?
The SuperJump protocol is quite intricate. What was the rationale that the authors used to establish the specific exercises and recovery phases? A description of this rationale is required. How do the authors define the active recovery phase?
The study will immensely benefit if compared with another control group that performs other forms of comparable physical activity than a control group that does not perform physical activity at all. The authors could perform a meta-analysis of publicly available data for GLP1, GIP, glucose, and insulin levels in individuals performing moderate exercise and compare that with the SuperJump protocol and include this in their discussion. The clear distinction there would be how the indoor SuperJump training still compares with outdoor physical activity.
GLP-1 is known to stimulate insulin release. Therefore, the observation of elevated active GLP-1, but reduced insulin is counterintuitive. How can the authors explain this contradictory result? An in-depth discussion of this is necessary.
The y axis for Fig 4B needs to be corrected as normal insulin levels are not expressed in micro IU/L but milli IU/L.
Author Response
A link between exercise and gut peptide production is well established. How does the SuperJump protocol differ from other forms of exercise in its ability to induce gut peptide production?
Thank you very much for this interesting point of discussion. The difference is that previous studies have focused on the effects of acute exercise on secretion of gastrointestinal hormones while in this study was analyzed the effects of chronic exercise (20 weeks of Superjump exercise) on endogenous peptides. The point has been clarified better in the discussion (lines 323-330).
The SuperJump protocol is quite intricate. What was the rationale that the authors used to establish the specific exercises and recovery phases? A description of this rationale is required. How do the authors define the active recovery phase?
Thank you for your suggestion. The performance and the effects of Superjump were recently studied (Iannaccone, 2020, Sustainability. 12(23): p. 1-10.; Vasto, 2022, Biol Sport, 2022. 39: p.1-6) The rationale was to do both resistance exercises and impact activities to exert osteogenic effects. Indeed, it’s demonstrated that resistance exercise has a better osteogenic potential than just aerobic exercise (Proia 2021, Front Endocrinol. 2021. 12: p. 704647.2021). The active recovery phase was important to prolong “the impact stimulus” of the activity over time. The impact of a physical exercise is the combination of force magnitude and the speed at which the force is applied (Gunter, 2012 Exerc Sport Sci Rev. 2012 40:13-21.). Activities with the most osteogenic potential have ground reaction forces (GRF) greater than 3.5 times BW (per leg), with peak force occurring in less than 0.1 s (Hind, 2007, Bone. 2007; 40:14Y27). Comparing three main activities such as walking, running, jumping the last has the greatest benefits to bone mineralization (MacKelvie, 2003 and 2004, Pediatrics. 2003; 112(6 Pt 1):e447, Bone. 2004; 34(4):755Y64). It also seems important not only the characteristic of the movement but also the number of repetitions, in fact 50 jumps in a session (Wiebe, 2008, Pediatr. Exerc. Sci. 2008; 20:211Y28.) have no osteogenic effect compared to 100 jumps (Fuchs ,2001 Bone Miner. Res. 2001; 16:148Y56.; Gunter, 2008, Bone. 2008; 42:710Y8.). Therefore, during the active recovery phase, the subjects of the exercise group continued to jump on the trampoline at the minimum intensity that allowed them to perform the jump (just lift both feet off the trampoline together). A description of the rationale with all this information, was added into the new section workout characteristics (lines 105-136)
- The study will immensely benefit if compared with another control group that performs other forms of comparable physical activity than a control group that does not perform physical activity at all. The authors could perform a meta-analysis of publicly available data for GLP1, GIP, glucose, and insulin levels in individuals performing moderate exercise and compare that with the SuperJump protocol and include this in their discussion. The clear distinction there would be how the indoor SuperJump training still compares with outdoor physical activity.
Thank you very much for your suggestion. We tried to do the meta-analysis of publicly available data as you suggested: About GLP-1 the study presented in literature has measured total GLP-1 and not active GLP-1 like in our study so it is not possible to compare the study. About GIP the studies have been conducted in obese/diabetic cohort of patients or following a glucose tolerance test. Moreover, both GLP-1 and GIP study were conducted in acute and not after several weeks of physical activity. We added the points to the discussion (lines 330-337)
GLP-1 is known to stimulate insulin release. Therefore, the observation of elevated active GLP-1, but reduced insulin is counterintuitive. How can the authors explain this contradictory result? An in-depth discussion of this is necessary.
Thank you for the suggestion. Yes, you are right. The observation of elevated GLP-1 and reduced insulin could be counterintuitive in consideration of the ability of GLP-1 to stimulate insulin release. Thus, it is necessary to point out that the incretin effect is defined as the increase in insulin response after an oral ingestion of glucose. In fact, GLP-1 induces insulin secretion via the GLP-1R in a glucose-regulated manner. It is possible to hypothesize that fasting levels of insulin and glucose were lower thanks to a regulatory mechanism of the peptide on beta-cells that could be sensitized to secrete the minimum amount of insulin required to have an accurate glycemic control. As you required the result was explained in depth into the discussion (lines 314-322).
- The y axis for Fig 4B needs to be corrected as normal insulin levels are not expressed in micro IU/L but milli IU/L.
done
Round 2
Reviewer 3 Report
The authors have provided precise and convincing answers to satisfy all my queries. I have no further comments.
Author Response
Thank you